# Challenges and Opportunities of Male Partner Involvement in Cervical Cancer Prevention and Control in Central Kenya: A Qualitative Analysis

**DOI:** 10.3390/ijerph22101575

**Published:** 2025-10-15

**Authors:** John H. Mwangi, Pretty N. Mbeje, Gloria N. Mtshali

**Affiliations:** School of Nursing and Public Health, University of KwaZulu-Natal, Durban 4000, South Africa; mbejep@ukzn.ac.za (P.N.M.); mtshalin3@ukzn.ac.za (G.N.M.)

**Keywords:** challenges and opportunities, male partner involvement, cervical cancer, prevention and control, qualitative analysis

## Abstract

**Background:** Cervical cancer remains a significant public health concern in Kenya, with male partner involvement increasingly recognized as a key factor in effective prevention and control. However, limited research has explored the specific barriers and enablers to such involvement in the Kenyan context. This study aimed to examine the challenges and opportunities associated with male partner involvement in cervical cancer prevention in Central Kenya. **Methods:** A qualitative descriptive design was employed. Purposive sampling was used to select 73 participants, including 20 couples (40 individuals), 20 nurses, 2 clinical officers, 2 gynecologists, 6 community health workers, and 3 county health directors. Data were collected through interviews and focus group discussions and analyzed thematically to identify key patterns and insights. **Results:** The mean age of male partners was 36.9 years, and 30.5 years for female partners. Most couples (70%, n = 28) had attained secondary education. The average duration of professional experience for nurses and clinical officers was 13 years. Key challenges included knowledge gaps, financial and logistical barriers, limited moral support, time constraints, sociocultural beliefs, stigma, and inadequate facility infrastructure. Identified opportunities included community education, shifting norms, improved couple communication, practical support from men, and integration of services. **Conclusions:** This study identifies key challenges and practical opportunities for increasing male involvement in cervical cancer prevention. Clarifying and promoting specific male roles such as support for screening and vaccination can enhance the effectiveness of cervical cancer prevention strategies in Central Kenya.

## 1. Introduction

Cervical cancer (CC) is one of the major public health problems globally, being the most frequent and fatal neo-plastic disease in the female population between 25 and 64 years. The global incidence was 662,044 in 2022 with 348,709 deaths [1]. In Kenya, CC is the second most common female cancer and the leading cause of cancer-related death among women aged (15–44), in 2023 there were 3591 cancer deaths with 5845 reported new cases [2]. Most morbidity and mortalities associated with CC occur in women in Sub-Saharan Africa, highest in east and southern Africa, and women living in low- and middle-income 206 countries (LMICs) bear 90% of the CC mortality [3].

CC affects women in their reproductive years, when they are caregivers and economic providers for their families. The Kenya 2017–2022 national cancer control strategy has identified prevention, screening and early detection as the first of five strategic objectives to cancer control [4]. The 2018 CC screening guidelines also proposes to have every woman age 25–49 screened regularly for cervical cancer [5]. Despite the well-established evidence on the effectiveness increased screening to reduce CC burden, the screening coverage and treatment of screen positive lesions remain low [6]. In order achieve the WHO 90–70–90 targets for CC elimination, which includes vaccination, screening, and treatment, the adoption of innovative strategies is crucial. Key strategies should extend beyond biomedical interventions to include behavioral approaches, such as promoting male partner involvement, given their significant influence on women’s health seeking behaviors and uptake of health services [7,8].

Male partner involvement in cervical cancer prevention is not merely a matter of personal responsibility; it is a critical component of a comprehensive public health strategy. Recognizing the interconnectedness of health within relationships, integrating male partners into prevention efforts can significantly enhance program effectiveness and reach [9]. This approach aligns with the principles of gender equality and promotes a more inclusive and holistic understanding of cervical cancer prevention. Male partners are critical decision makers in family affairs, including reproductive healthcare matters, yet the majority of programs almost exclusively focus on the reproductive health behavior of women, with men mostly being marginalized. Men have a critical role to play in the reproductive health of themselves, their partners, and the family as a whole [10]. Male partner involvement in cervical cancer prevention has been shown to include supporting women’s attendance at cervical screening services, encouraging and facilitating HPV vaccination for daughters, adopting responsible sexual behavior to reduce HPV transmission, providing financial and logistical support for preventive services, offering emotional and moral support, and participating in health education and community awareness initiatives [11,12]. However, cultural and social norms, and limited knowledge of CC impede male involvement [13].

Opportunities for male involvement in CC prevention and control are significant. Previous studies report improved maternal, child and reproductive health outcomes due to male involvement [14]. However, little is known about how male involvement can be effectively promoted in settings where entrenched unequal gender roles, norms and relations constrain women from effectively inviting men to participate in reproductive health [15].

Involving men in reproductive healthcare could help in achieving some major development goals, such as a decreased maternal mortality rate and an increased contraceptive prevalence rate. Involving men could also help reduce the overall prevalence of HIV/AIDS—an outcome possible only if men are involved not just as clients of reproductive healthcare but also as partners, service providers, policy makers, teachers, and project managers. While the benefits of male involvement have been acknowledged, there continues to be a challenge in creating a space for and engaging men in maternal and reproductive health. Given that men are important as partners, fathers and healthcare professionals, it is important to involve and engage with them in reproductive health issues [16].

Despite the critical role that male partners can play in alleviating the burden of cervical cancer, there is minimal information on their involvement in the screening and treatment process including challenges and opportunities [17]. Understanding these perspectives through a qualitative study is crucial in providing contextual understanding of social dynamics, and culturally sensitive interventions that could promote male involvement in cervical cancer prevention and treatment.

Spousal support in health matters has a positive impact on health promotion and the mitigation of ill health [18]. Identifying the challenges and potential opportunities of male partner involvement in the uptake of cervical cancer screening and treatment services will enhance the implementation of programs at the facility, county, and country levels in Kenya. This improvement is expected to enhance the quality and delivery of health services.

This study qualitatively explores the challenges and opportunities of male partner involvement in CC prevention and control in Kenya through the lens of the social ecological model (SEM). Understanding challenges and opportunities of male partner involvement is crucial in guiding interventions that seek to strengthen male engagement, which could potentially improve the uptake and effectiveness of CC prevention and control strategies in Kenya.

## 2. Materials and Methods

### 2.1. Study Design

This study employed a qualitative, analytical descriptive approach with Interpretivism and Constructivism ideologies. This approach advocates the notion that individuals are intentional and inventive in their actions, actively shaping their social environment. The approach recognizes the dynamic and evolving nature of society, acknowledging the possibility of multiple interpretations of an event influenced by individuals’ historical or social perspectives. The relevance of the approach to this study lies in the historical focus of reproductive health programs primarily on women as clients, overlooking the fact that women are not entirely independent of men. Women often have male figures in their lives, such as husbands, fathers, male relatives, and other significant others. This study was conducted between 8 April 2024 and 7 June 2024.

### 2.2. Study Setting

This study was conducted in three county referral hospitals, MCH clinics and community settings in Central Kenya, namely Murang’a, Nyeri, and Kirinyaga. The sites were purposefully selected based on the prevalence of cervical cancer as depicted by hospital-based registries and availability of cervical cancer screening and treatment services.

### 2.3. Population and Sampling

The study population included adult heterosexual couples (where the female partners was of reproductive age), frontline healthcare providers (nurses, clinical officers, and gynecologists), community health workers (CHWs), and county health officials involved in cervical cancer programming. Purposive sampling was used to select a diverse range of participants who could offer different perspectives and experiences and who were comfortable providing the information needed for this study. Couples defined (men and women of reproductive age and in an intimate relationship) were eligible for the interviews if they were of reproductive age and willing to provide consent, healthcare workers were eligible if they had prior training on cervical cancer screening and treatment and had worked in the clinics for at least six months, community health workers were stationed within local communities, gynecologists and county health directors were deployed in the local counties.

### 2.4. Inclusion Criteria

Partners who attended maternal and child health clinics in the study area and provided written consent to participate in the focused group discussion. Nurses and clinical officers who were trained on cervical cancer screening and had worked for at least six months in the MCH clinic and were willing to participate in this study. Community health workers and county health directors who were available at their work stations and were willing to participate in this study.

### 2.5. Exclusion Criteria

Participants in the study area who declined to provide consent for participation in this study. Nurses and clinical officers who had worked in the cervical cancer screening area for less than six months.

### 2.6. Research Instruments

Data were collected using a semi-structured face-to-face interview schedule with open-ended questions to explore participants’ experience in relation to male partner involvement in cervical cancer prevention and control. Audio recording devices, pen and papers were part of research instruments.

### 2.7. Data Collection

Individual face-to-face interviews using a semi-structured guide were used to collect qualitative data. The interviews were conducted at a time convenient for each participant and each interview lasted between 30 and 45 min. This study involved a total of 20 couples of reproductive age (20 males and 20 females), and IDIs with providers and policy makers (20 nurses and 2 clinical officers who had been trained in cervical cancer screening and had worked in the clinics for at least 6 months, 6 community health workers, 2 gynecologists and 3 county directors of health who represented policy makers). Field notes and audio recordings were collected for all the interviews. Focus group discussion involved 20 couples who were recruited at designated health facilities. FGDs were moderated by a research scientist who did not have any clinical roles in the designated facilities and provided a neutral environment for discussion. A note taker aided the moderator throughout the FGD session by documenting notes. FGDs were conducted in participants’ preferred language of Swahili. FGDs were translated verbatim into English with experienced research assistants. In-depth interviews (IDIs) with providers and policy makers were conducted in English. Interviews were conducted in a private space with the participant and the interviewer only. Interview participants were not provided with question bank prep prior to the interviews to minimize bias that could arise from structured responses. Data saturation was determined within each participant category by identifying repetition of themes and no emergence of new information during data collection and preliminary analysis. Participants’ personal information and data collected were kept confidential. All recordings and transcriptions were saved in a password protected computer. During data collection, participants were not provided with incentive reimbursement; however, light refreshments were offered during the sessions and return fare was also provided. This approach ensured that participation was voluntary and not unduly influenced by monetary incentives, which are in line with established ethical principles for research involving human subjects.

### 2.8. Data Analysis and Interpretation

Data were analyzed using inductive and deductive thematic analysis approaches. The following steps were taken in the analytical process: verbatim transcription of the audios and field notes into Microsoft Word; confirming and validating the transcripts; putting together the thematic framework; coding of the recorded conversations using the thematic framework; and charting and interpreting the data [19]. Emerging themes were recorded throughout the transcription process. Following transcriptions, several readings were conducted in order to internalize and pinpoint particular ideas that were then further expounded upon. The first step involved familiarization with the data by going through all the data scripts individually, by the researcher and assistants to resolve any existing discrepancies in the codes. And once the codes were agreed upon the two analysts reviewed and revised the codebook (Appendix A), and coding was conducted independently. Coding was supported by the Dedoose software (Version 10.0.35, https://www.dedoose.com/ (accessed on 20 August 2025)) Thematic analysis involved identifying recurring themes, topics, ideas, patterns, and meaningful categories that emerge from the data collected. Different topics were examined, followed by the establishment of themes and compilation of a summary of the results. The qualitative methods adhered to the consolidated criteria for reporting qualitative research [20]. Participants’ personal information and data collected were kept confidential.

### 2.9. Trustworthiness of This Study

The criteria for guaranteeing trustworthiness, i.e., transferability, authenticity, confirmability, dependability and credibility, were satisfied [21]. These elements help validate the accuracy of the research process and the integrity of its findings. After the data were transcribed, an independent coder was involved to confirm authenticity. The results of this study were thoroughly described after a thematic analysis of the data. Transferability was ensured by a detailed presentation of the study results. Data audio recording and use of direct quotes from the participants ensured authenticity. Confirmability and credibility were assessed by member checking and peer debriefing to ensure that data accurately reflected the participant experiences. A committed and proficient research assistants ensured data dependability by their consistency in data collection and audit trail of data analysis.

### 2.10. Ethical Consideration

The initial step involved obtaining research approval from the University of KwaZulu-Natal’s Biomedical Research Ethics Committee (BREC), reference number BREC/00006580/2023. Additional ethical clearance from the country in which this study is based was secured from Mt. Kenya University’s Ethics Review Committee, reference number MKU/ISERC/3433. A license to conduct the research was granted by the National Commission for Science, Technology, and Innovation (NACOSTI) in Kenya, reference number 495117. Authority to collect data was also obtained from the relevant County Government Health offices in the study areas. Data collection commenced after securing written and signed informed consent from the participants, who had the right to withdraw at any time. Privacy and confidentiality were upheld, ensuring that data could not be traced back to individual participants. In all the stages of this study, the ethical principles were upheld that included respect for persons, beneficence, integrity, objectivity and justice.

## 3. Findings

### 3.1. Characteristics of the Participants

Overall, we interviewed 73 participants—33 key informants and 40 FGD participants. Among the key informants, the majority were nurses (61%), community health workers (18%), county directors of health (9%), clinical officers (6%), and gynecologists (6%), with more females (20) compared to males (13). FGDs achieved gender parity with 20 males and 20 females. The majority of the key informants had diplomas (58%), while others had degrees or higher qualifications such as master’s degrees. A small proportion, particularly community health workers, had only high school or primary school level education. Among FGD participants, majority had attained high school level education (70%), while fewer had progressed to college (15%) or had only primary school education (15%). Age and experience profiles also differed between the groups. Key informants were generally older, with a mean age of 41.1 years (range 23–55) compared to a mean of 33.7 years (range 21–50) among FGD participants. The median ages were 42.0 and 34.5 years, respectively. Work experience data were collected for key informants only, showing an average of 12.8 years (range 1–32 years) with a median of 12 years, indicating substantial professional experience within the groups shown in Table 1. The participants were labelled as N_n_, CO_n_, CHW_n_, G_n_, CDH_n_, and FGD1-3_yn or xn_, where N stands for nurse, CO for clinical officer, CHW for community health worker, G for gynecologist, CDH for county director of health, FGD1-3 for focus group discussion one to three, n for serial number, x for female member, and y for male member.

### 3.2. Thematic Analysis from Qualitative Data and Emerging Themes

Qualitative data findings from focused group discussions with couples and interview schedules with community health workers, healthcare providers and county directors of health were analyzed and organized into themes and subthemes. These were further classified into various levels based on social ecological model. This is illustrated in Table 2.

The subthemes that emerged from the participants’ discussion and interviews were aligned according to the social ecological model to reflect various levels (individual factors, interpersonal, in community, and structural factors) as illustrated in Figure 1.

#### 3.2.1. Theme 1: Individual Factors

Male partner barriers and opportunities related to individual knowledge, perceptions, and motivation male involvement in cervical cancer prevention and control.

##### Lack of Understanding on Causes and Progression of Cervical Cancer

Some of the community health workers reported that the majority of men in their area of service lacked knowledge about cervical cancer. A common challenge across all participant groups was the limited understanding of cervical cancer, its transmission, and prevention. Both men and women demonstrated low awareness of HPV and its link to cervical cancer. This knowledge gap contributed to fear, misconceptions, and poor uptake of screening services. Several participants believed that cervical cancer was not sexually transmitted, while others associated the disease with curses or fate. Health workers also noted that men often lacked basic information about the disease, including the availability of screening and vaccination services.


*“Majority of men here are ignorant on the issue about cervical cancer, and some think women who get diagnosed with it are cursed”*
(CHW_4_)

Similarly, there was widespread distrust or misunderstanding of the HPV vaccine. Some male participants were reluctant to have their daughters vaccinated due to fears about side effects, misinformation, or belief that cervical cancer does not affect young girls.


*“I don’t trust this vaccine they are giving our girls at school; small girls cannot get cervical cancer.”*
(FGD1_Y6_)

##### Lack of Awareness of Prevention and Control Measures

Some of the focused group discussion members reported lack of awareness on where the cervical cancer screening services are offered and the cost implicated.


*“What about cervical cancer screening services? Is it free in all public hospitals in Kenya?”*
(FGD1_y1_)

The researcher and his assistants elaborated on the process of screening and where the services can be accessed. Some of the nurses also explained how, many of her clients were ignorant on basic facts about cervical cancer.


*“Many clients coming to our clinics have no basic facts regarding cervical cancer and its prevention”*
(N_10_)

More than half of the focused group discussion members were not aware that cervical cancer can be transmitted sexually.

“*Cancer is not an infection how can it be transmitted sexually*”(FGD1_Y6_)


*“Do you mean those women with cervical cancer got it from their partners?”*
(FGD1_X4_)

The investigator clarified how HPV is associated with cervical cancer. One of the nurse participants also stated that many men were not aware cervical cancer can be sexually transmitted.


*“Majority of men are not even aware that Cervical cancer can be sexually transmitted”*
(N_13_) 

#### 3.2.2. Theme 2: Interpersonal (Intrapersonal) Factors

Dynamics within families and relationships that shape male involvement.

##### Lack of Moral Support and Poor Couple Communication

Participants frequently reported poor communication and limited emotional support between spouses regarding reproductive health. Men often showed disinterest in cervical cancer, viewing it as a “women’s issue.”


*“I would appreciate if my husband accompanies me to the clinics, but he always says there is no need as I am not sick.”*
(FGD2_X2_)

Healthcare workers supported this perception, stating that men rarely participated in discussions about their partners’ health.


*“There is lack of effective communication between women and their spouses. Majority do not even inform their partners when coming to the clinic.”*
(N_12_)


*“Men will only come to hospital when they are severely sick. Screening services are alien to them.”*
(N_14_)

##### Competing Responsibilities and Lack of Motivation

Lack of time due to work commitments and prioritization of economic activities also limited male participation. Many women in focus groups reported that their male partners were unavailable due to work obligations, and therefore they often attended hospital visits alone.


*“Mine complains of lack of time; he says he is busy looking for money and I am also hustling.”*
(FGD2_X6_)


*“Most men are busy looking for money and drinking, so they are not readily available.”*
(N_16_)

Additionally, long waiting times and frequent service interruptions were cited as deterrents for male participation.


*“Some males who come usually complain of slow-moving queues and nurses going for tea breaks before they are served.”*
(N_18_)

Healthcare workers also noted that men often deprioritized preventive care.


*“Men do not prioritize taking their women for cervical cancer procedures. It is not urgent in their eyes.”*
(N_7_)

##### Opportunities for Couple Communication and Joint Decision Making

Open dialogue and shared decisions were seen as pathways to stronger involvement. Effective communication was seen as lacking but necessary for engagement.


*“There is lack of effective communication between women and their spouses as majority do not even inform their partners when coming to the clinic.”*
(N_12_)


*“Men usually come to the clinic when they are informed that there will be no sexual intercourse for six weeks post cryotherapy to ascertain the information.”*
(N_2_)

Importantly, the need for joint decision making was highlighted as crucial, which would promote knowledge and improve support offered by male partners.


*“Men need to understand their roles in their partners’ health… because in case they are diagnosed with cervical cancer, everyone in the family will be affected.”*
(CHW_5_)


*“When we are giving health education to couples within our designated areas of service, we usually see men eager to get information regarding reproductive health issues.”*
 (CHW_6_)

#### 3.2.3. Theme 3: Community Factors

Norms, cultural beliefs, and leadership influence male participation.

##### Social Cultural Norms and Traditions

Cultural transformation was seen as critical to changing male attitudes and behaviors.

Participants highlighted the need to redefine masculinity. Reframing male roles to include support for women’s health was suggested as a transformative strategy.


*“From my experience, women with supportive partners are screened more and they usually complete their follow-up schedules as opposed to those whose men aren’t concerned.”*
(N_7_) 

Myths about male presence in women’s reproductive health services need to be challenged.


*“We don’t usually allow men to witness their partners being screened for cervical cancer.”*
(N_9_)


*(When probed) “It’s local community culture not to allow men when women are being ‘exposed’.”*
(N_9_)

##### Stigma and Fear

Stigma or negative perception or social disapproval associated with a particular characteristics or behavior in this case being associated with cervical cancer was reported as one of the challenges.

Fear of Screening Procedures and Outcomes: Some Male partners avoid involvement due to fear of diagnosis or witnessing screening.


*“Some women say their spouses fear the screening or being informed the results are positive.”*
(N_17_)

Fear of Being Seen at Maternal Clinics: Social stigma and perceived loss of masculinity prevent men from accompanying partners.


*“Some men are embarrassed to accompany their wives to maternal and child health clinics.”*


##### Role of Community and Religious Leaders

Local leaders sometimes spread misinformation, discouraging HPV vaccination or Pap smears.


*“Some community leaders discourage participation in HPV vaccination and Pap smear services.”*
(N_4_)


*“Some pastors advise parents not to vaccinate their girls, claiming it causes sterility.”*
(CO_1_)

At the same time, leaders and influencers were identified as potential allies.


*“When the leaders including ward reps, chiefs, headmen, teachers and others lead by example… other members of the community will likely do the same.”*
(CO_1_)


*“Priests and imams are highly respected members of the community. If they are advised accordingly, they can help to pass health messages related to cervical cancer.”*
(CO_2_)

#### 3.2.4. Theme 4: Structural (Health System and Policy) Factors

Institutional, economic, and policy-level factors shaping male involvement.

##### Financial and Logistical Barriers

Economic hardship limited access to screening and follow-up.


*“The other day they examined me and said I should go for further tests. I didn’t have the money, since we had paid fees for our son in college.”*
(FGD1_X3_)

Subsistence farming and poverty were common obstacles.


*“Majority of community members are financially poor and are subsistence farmers. When rain fails, they don’t even have food.”*
(N_9_)

Logistical barriers such as lack of transport and childcare also hindered women’s attendance.


*“Some of my friends can’t attend these clinics you are talking about, as they are overwhelmed by household chores including child-care and farm work.”*
(FGD2_X1_)

##### Health System and Infrastructure Barriers

Participants reported distrust in public services, shortage of male-friendly spaces, and lack of privacy.


*“Some nurses take their children to private clinics—does it mean they don’t trust their own services?”*
(FGD1_X1_)


*“The sitting arrangement is so squeezed—do you expect us to fight for seats with pregnant women and children?”*
(FGD1_Y4_)

The predominance of female staff was cited as a deterrent.


*“The ratio of female to male staff is high, which may discourage Male partners from coming to the clinic.”*
(N_19_)

##### Opportunities Through Policy and Service Integration

Participants highlighted the need for male-friendly spaces, service integration, and incentives.

“*We can offer male-friendly services like PSA screening to encourage men to be accompanying their spouses.”*
(N_5_)


*“We can also advise KEPI to include HPV vaccine in the routine childhood vaccines so that we can reduce missed opportunities.”*
(CHD_3_)

Healthcare leaders emphasized system-level change:


*“There is no policy that can directly force males to participate… the only thing we can do is to make the hospital or clinic environment accommodative or friendly to them.”*
(CHD_3_)

## 4. Discussion

This study explored the challenges and opportunities for male involvement in cervical cancer prevention and control in public health facilities in Central Kenya. The findings of this study describe male involvement in cervical cancer through the lens of social ecological model (SEM) model, including individual, interpersonal, community and structural factors. At the individual level, both men and women demonstrated limited awareness of human papillomavirus (HPV) as the primary cause of cervical cancer, with misconceptions framing the disease as a curse, fate, or unrelated to sexual transmission. Such misinformation contributed to fear, stigma, and poor uptake of cervical screening and vaccination services. At the interpersonal level, gaps in male partners’ knowledge not only undermined their own engagement but also limited their capacity to support women in seeking preventive services. For instance, reluctance to vaccinate daughters against HPV stemmed from distrust, misinformation, and beliefs that cervical cancer does not affect young girls. These findings underscore how individual and relational knowledge gaps contribute to barriers in cervical cancer prevention and control, while emphasizing the need for culturally tailored educational interventions that address both men and women within the household and community context.

Awareness campaigns and education was a key opportunity for male involvement. First, education and awareness campaigns targeting men can dispel myths and improve knowledge. Workshops, peer education, and early integration of sexual health into school curricula can shift perspectives and promote male advocacy [23]. Notably, a study in Nigeria showed high willingness of men in supporting CC screening and prevention among their female partners, which underscores the role of male involvement [24]. Outreach services that bring screening closer to communities can also improve access. Secondly, shifting social norms can help redefine masculinity to include support for women’s health. Engaging men as equal partners in reproductive health discussions and empowering them to encourage their spouses fosters shared responsibility [25]. Community-wide sensitization can demystify taboos and promote gender-equitable attitudes. Effective communication between partners is also vital. Encouraging open dialogue about sexual health, screening, and vaccination enhances mutual understanding and informed decision making. These findings support similar studies advocating for couple-based approaches in reproductive health [26,27]. Additionally, Male partners can be involved through practical support, including accompanying partners to clinics, helping with household chores, and providing emotional and logistical support during treatment [28]. An additional critical opportunity for strengthening prevention lies in the integration of health services. Incorporating male-oriented health checks, such as prostate-specific antigen (PSA) screening, alongside cervical cancer prevention programs could promote couple-centered engagement and joint participation in cancer prevention. Furthermore, embedding HPV vaccination within the Kenya Expanded Programme on Immunization (KEPI) offers a strategic pathway to increase vaccine coverage by reaching both girls and boys prior to sexual debut, thereby enhancing the long-term effectiveness of cervical cancer prevention efforts [29]. Facility environments also need to be improved by expanding infrastructure, including more male healthcare workers in MCH clinics, and streamlining services to reduce waiting times.

Lastly, engaging role models—including community leaders, religious figures, and media influencers—can normalize male participation and encourage others to follow suit. Leveraging digital platforms and mobile networks to disseminate tailored messages on cervical cancer can further expand reach [30,31]. Role modeling by respected community figures can help counter stigma and foster community acceptance.

We also report key barriers to male involvement in cervical cancer prevention and treatment. A key individual-level barrier was limited knowledge. Many men lacked awareness about the causes and progression of cervical cancer, the importance of screening, and the safety of HPV vaccination. This was often rooted in myths, misinformation, and the cultural taboos that inhibit open discussions around sexual and reproductive health [32,33]. Such knowledge gaps diminish men’s capacity to support their partners in seeking preventive services. Similarly, a qualitative study conducted in Uganda on involving men in CC prevention also found limited knowledge among men, with existing misinformation about risk factors and causes of cancer [34].

Financial and logistical barriers also played a major role. With limited household incomes, reproductive health was often deprioritized. The inability to support transport, pay for diagnostic tests, or hire help for domestic chores discouraged women from accessing screening services [35]. Moreover, the lack of emotional and moral support from male partners—manifested through poor spousal communication, disinterest in cervical cancer, and fears about infidelity—further hampered women’s health-seeking behavior. This aligns with prior studies [36,37] showing men often perceive reproductive health as solely a woman’s responsibility.

Time constraints were another major hindrance. Many men cited busy work schedules and long waiting times at clinics as deterrents. The inflexibility of clinic hours and the absence of male-friendly services made attendance more difficult, contributing to disengagement [38,39]. Social-cultural norms also restricted involvement. Traditional views of masculinity, stigma around discussing reproductive health, and fears of being judged for supporting female health services reinforced male partner detachment [40].

Health facility factors further discouraged involvement. Participants noted poor infrastructure, lack of privacy, female-dominated staff in MCH clinics, and exclusion of men during cervical procedures. These factors made health environments unwelcoming to male partners [41]. Stigma was another layer of resistance. Some men feared ridicule or judgement for accompanying their partners, while others feared the emotional burden of a cancer diagnosis [42]. In addition, a lack of proactive community leadership weakened efforts to mobilize male involvement. Religious and traditional leaders were either silent or spread misinformation—especially regarding the HPV vaccine—thus discouraging community participation [43].

## 5. Strengths and Limitations of This Study

An important strength of our study is the use of qualitative approaches, which enabled a deeper and more nuanced understanding of male involvement in cervical cancer treatment and prevention. A key limitation, however, is the potential for social desirability bias due to the dynamics of focus group discussions. Additionally, our recruitment targeted couples accessing health services, and their perspectives may differ from those of other populations. Despite these limitations, this study provides an in-depth understanding of the barriers to, and opportunities for, male partner involvement in cervical cancer treatment and prevention.

## 6. Conclusions

In conclusion, this study used a qualitative approach to understand multifaceted barriers and opportunities related to male involvement in CC prevention and control. We found a complex interplay of individual, interpersonal, community, and structural factors that influence male involvement in cervical cancer prevention and control. Specifically, male involvement in CC prevention and control is hindered by various factors. At the individual level, there is a significant lack of knowledge and misconceptions about CC, its causes, transmission, and prevention, which contributes to fear and distrust, especially concerning the HPV vaccine. Interpersonal dynamics also impede male participation, with poor communication, limited emotional support, and competing responsibilities reducing men’s engagement in reproductive health matters. Community norms and cultural beliefs reinforce male disengagement through stigma, fear, and restrictive cultural traditions, while misinformation among community and religious leaders who are expected to provide direction is also challenging. Structural factors such as financial constraints, logistical challenges, distrust in health services, and lack of male-friendly spaces further limit access and involvement. Importantly, opportunities exist through enhancing couple communication, redefining male roles, leveraging community leadership positively, and implementing health system reforms that create male-friendly environments and integrate services. Addressing these multi-level factors holistically is essential to improve male participation and, consequently, the effectiveness of cervical cancer prevention and control efforts.

## Figures and Tables

**Figure 1 ijerph-22-01575-f001:**
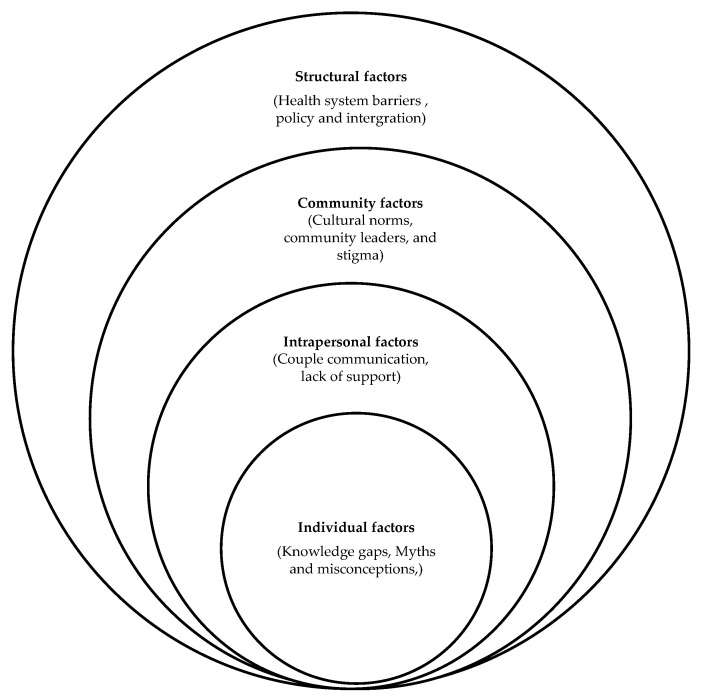
Socio-Ecological Model Illustrating Factors Influencing Male Partner Involvement in Cervical Cancer Prevention and Control [22].

**Table 1 ijerph-22-01575-t001:** Study sites and characteristics of participants in the IDIs and FGDs.

Characteristic	Key Informants (n = 33)	FGD Participants (n = 40)
Participants	33	40
Gender: Female	20	20
Gender: Male	13	20
Facility: Murang’a	12	14
Facility: Nyeri	12	14
Facility: Kerugoya	9	12
Category: Nurse	20	–
Category: CHW	6	–
Category: County Director	3	–
Category: Clinical Officer	2	–
Category: Gynecologist	2	–
Education/Qualification:		
Diploma	19	–
Degree	6	–
Masters	2	–
High school	4	–
Primary school	2	–
Secondary school	–	28
College	–	6
Primary school	–	6
Age (years): Mean (Range)	41.1 (23–55)	33.7 (21–50)
Age (years): Median	42.0	34.5
Experience (years): Mean (Range)	12.8 (1–32)	–
Experience (years): Median	12.0	–

**Table 2 ijerph-22-01575-t002:** Thematic analysis.

SEM Level	Themes/Subthemes	Illustrative Quotes
**Individual Factors**	Lack of understanding of causes and progression of cervical cancer Myths and misconceptions about HPV vaccine Lack of awareness of prevention and control measures	*“Majority of men here are ignorant on the issue about cervical cancer, and some think women who get diagnosed with it are cursed*.” (CHW_4_) *“I don’t trust this vaccine they are giving our girls at school; small girls cannot get cervical cancer.”* (FGD1_Y6_) *“Many clients coming to our clinics have no basic facts regarding cervical cancer and its prevention.”* (N_10_)
**Interpersonal Factors**	Lack of moral support and poor couple communication Competing responsibilities and lack of motivation Opportunities for open dialogue and shared decision making	*“I would appreciate if my husband accompanied me to the clinics, but he always says there is no need as I am not sick.”* (FGD2_X2_) “*Mine complains of lack of time; he says he is busy looking for money and I am also hustling.”* (FGD2_X6_) *“Men need to understand their roles in their partners’ health… because in case they are diagnosed with cervical cancer, everyone in the family will* *be affected.”* (CHW_5_)
**Community Factors**	Social cultural norms and traditions discouraging male involvement Stigma and fear of screening/being seen at maternal clinics Influence of community and religious leaders (both misinformation and opportunities for engagement)	*“It is difficult for men to accompany their wives here due to customs and traditional perceptions.”* (N_17_) *“Some men are embarrassed to accompany their wives to maternal and child health clinics.” (N_5_) “Some pastors advise parents not to vaccinate their girls, claiming it causes sterility.”* (CO_1_) *“When the leaders including ward reps, chiefs, headmen, teachers and others lead by example… other members of the community will likely do* *the same.”* (CO_1_)
**Structural (Health System and Policy) Factors**	Financial barriers (costs of tests, transport, poverty) Logistical barriers (childcare, transport, workload)	*“The other day they examined me and said I should go for further tests.I didn’t have the money, since we had paid fees for our son in college”* (FGD1_x3_) *“some of my friends can’t attend these clinics you are talking about, as they are overwhelmed by houdehold chores including child-care and farm* work’’ (FGD2_X1_)
	Health system challenges (lack of trust, female-dominated staff, inadequate infrastructure, lack of privacy) Opportunities for policy reform and service integration (male-friendly clinics, HPV in KEPI, incentives)	*“The sitting arrangement is so squeezed-do you expect us to fight for seats with pregnant women and children?”* (FGD1_Y4_) *“There is no policy that can directly force males to participate… the only thing we can do is to make the hospital or clinic environment accommodative or friendly to them.”* (CHD_3_)

## Data Availability

The original contributions presented in this study are included in this article. Further inquiries can be directed to the corresponding author.

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
