# Peer review of "Challenges and Opportunities of Male Partner Involvement in Cervical Cancer Prevention and Control in Central Kenya: A Qualitative Analysis"

_ijerph, 2025, doi:10.3390/ijerph22101575_

Round 1

Reviewer 1 Report (Previous Reviewer 1)

Comments and Suggestions for Authors

For me, a hallmark of scientific research is its reproducibility. Based on this section, I would not know how to conduct a similar study in my own country. In my view, this study is not reproducible for the reader. Additionally, the authors did not make revisions to the manuscript according to the suggestions when they submitted the paper to Healthcare. In my opinion, the manuscript is unacceptable and, moreover, will not be of interest to the reader.

Comments on the Quality of English Language

I am not a native speaker. 

Author Response

  1. For me, a hallmark of scientific research is its reproducibility. Based on this section, I would not know how to conduct a similar study in my own country. In my view, this study is not reproducible for the reader. Additionally, the authors did not make revisions to the manuscript according to the suggestions when they submitted the paper to Healthcare. In my opinion, the manuscript is unacceptable and, moreover, will not be of interest to the reader.

We thank the reviewer for this observation we have revised the manuscript method section for clarity as also recommended by other reviewers.

Reviewer 2 Report (Previous Reviewer 2)

Comments and Suggestions for Authors

Dear Authors 

Thanks for your detailed responses.

Author Response

  1. Thanks for your detailed responses.

We thank the reviewer.

Reviewer 3 Report (Previous Reviewer 3)

Comments and Suggestions for Authors
  • Clearly state the study’s novelty and position it against existing Kenya/SSA literature on male involvement in cervical cancer prevention. Add a stronger literature gap statement
  • Provide full methodological transparency following COREQ (recruitment process, refusals, FGD composition, moderator details, question bank prep, translations back and forth).
  • Include a codebook with theme development steps and example quotes linked to codes.
  • Clarify or remove mention of “triangulation with quantitative data” if not actually performed.
  • Deepen analysis by mapping themes to a conceptual framework for eg. socio-ecological model and linking individual, interpersonal and structural levels.
  • Support implementation recommendations such as PSA co-screening with feasibility and cost evidence, or frame them as something you would look at in future. Elaborate more.
  • Expand limitations to include social desirability bias, moderator influence, translation issues and possible bias from educational content during data collection
  • Add a table of participant per theme and site to improve transparency.
  • Indicate whether incentives were given to participants.
  • Add a visual schematic to summarize the study design and thematic relationships.

Author Response

  1. 1.Clearly state the study’s novelty and position it against existing Kenya/SSA literature on male involvement in cervical cancer prevention. Add a stronger literature gap statement

    We have included the literature gap and discussions on the Kenyan context in the introduction section. Line 68-72

    1. Provide full methodological transparency following COREQ (recruitment process, refusals, FGD composition, moderator details, question bank prep, translations back and forth).

    We have clarified and included details in the data collection section now reads “Focus group discussion involved 20 couples who were recruited at designated health facilities. FGDs were moderated by a research scientist who did not have any clinical roles in the designated facilities and provided a neutral environment for discussion. A note taker aided the moderator throughout the FGD session by documenting notes. FGDs were conducted in participants preferred language of Swahili. FGDs were translated verbatim into English with experienced research assistants. In-depth interviews (IDIs) with providers and policy makers were conducted in English. Interviews were conducted in private space with the participant and the interviewer only.  Interview participants were not provided with question bank prep prior to the interviews to minimize bias that could arise from structured responses”.

    3.Include a codebook with theme development steps and example quotes linked to codes.

    We have attached a codebook including linked quotes and appendix. And also included how we identified patterns and developed the codebook in the method section.

    4.Clarify or remove mention of “triangulation with quantitative data” if not actually performed.

    The Authors have done a quantitative study entitled “Male partner roles in cervical cancer transmission and prevention in central Kenya: A quantitative approach” which is published and can be accessed at https://doi.org/10.4102/hsag.v30i0.2858

    5.Deepen analysis by mapping themes to a conceptual framework for e.g. socio-ecological model and linking individual, interpersonal and structural levels.

    We thank the reviewer for this important note. We have included the social ecological model and revised the results to reflect individual, interpersonal, community and structural factors.

    6.Support implementation recommendations such as PSA co-screening with feasibility and cost evidence or frame them as something you would look at in future. Elaborate more.

    We have clarified this in the discussion lines 417-422 now reads. “An additional critical opportunity for strengthening prevention lies in the integration of health services. Incorporating male-oriented health checks, such as prostate-specific antigen (PSA) screening, alongside cervical cancer prevention programs could promote couple-centered engagement and joint participation in cancer prevention. Furthermore, embedding HPV vaccination within the Kenya Expanded programme on Immunization (KEPI) offers a strategic pathway to increase vaccine coverage by reaching both girls and boys prior to sexual debut, thereby enhancing the long-term effectiveness of cervical cancer prevention efforts”

    7.Expand limitations to include social desirability bias, moderator influence, translation issues and possible bias from educational content during data collection.

    8.We have included social desirability bias as a potential limitation for the study given the nature of data collection. Lines 454-459 now reads. “An important strength of our study is the use of qualitative approaches, which enabled a deeper and more nuanced understanding of male involvement in cervical cancer treatment and prevention. A key limitation, however, is the potential for social desirability bias due to the dynamics of focus group discussions. Additionally, our recruitment targeted couples accessing health services, and their perspectives may differ from those of other populations. Despite these limitations, the study provides an in-depth understanding of the barriers to, and opportunities for, male partner involvement in cervical cancer treatment and prevention”.

    9.Add a table of participant per theme and site to improve transparency.

    We have included table 2 in line 214 which includes the SEM constructs and the relevant quotations.

    10.Indicate whether incentives were given to participants.

    We have clarified this in lines 137-140 now reads “During data collection, participants were not provided with incentive reimbursement; however, light refreshments were offered during the sessions and return fare was also provided. This approach ensured that participation was voluntary and not unduly influenced by monetary incentives, which are in line with established ethical principles for research involving human subjects”.

    11.Add a visual schematic to summarize the study design and thematic relationships.

    We have provided the visual representation of our findings as in the social ecological model in (Figure 1)

Reviewer 4 Report (New Reviewer)

Comments and Suggestions for Authors

Overall this article offers a novel look into potential barriers to cervical cancer screenings in Central Kenya. The focus on the role of the male partner in attempting to increase cervical cancer screenings and other prevention methods is a fascinating project and, for this study, we see novel findings. While the results are interesting, I believe the total manuscript would benefit from additional revisions. The introduction, methods, and discussion could all be improved in various capacities. 

Introduction: Overall, it is simply too long of an introduction section. The total length is around 3 pages and must be cut. For example, lines 79-85 feel out of place in the introduction and would be better suited for the discussion. 

On lines 98-113 the authors go into great detail for a single article for one justification for the study. This can be greatly reduced in size or moved to the discussion but as it stands, this is not suitable. 

We see this in the following paragraph with lines 114-123. Great amounts of detail for one study. This should be adjusted. 

Authors should re-read the introduction to ensure the narrative is coherent and not using space to overexplain single articles. 

Finally, for the introduction, we do not receive information on why this should be a qualitative study outside of faint mention in lines 132-134. It would benefit the paper to expand on the rationale for the chosen methodology.

Methods: 

On line 168 the researchers discussion that "couples" were selected but does not define how they determine who is classified as a couple. Please add.

In the data analysis step, the way in which transcripts were coded and analyzed is discussed but we do not see how many reviewers were included. Please add. We see that in line 223-225 there were multiple reviewers involved but only at the end. Clarify.

Same thing for section 2.9, please specify the number of reviewers and coders assisting in this project.

Findings Section:

Table 1 and 2 are nice to have but I believe they are unnecessary. The same information could be conveyed in an additional paragraph in section 3.1.

Discussion: 

Currently, due to how long the introduction is, the discussion feels quite truncated. I do not believe much needs done as this can be adjusted with reducing the volume of the intro.

Limitations:

In general, the limitations section is rather bland and could use refinement. 

The methods section is quite strong in some regards but could use a bit of expansion in a couple areas. For inclusion criteria, lines 175-176, the manuscript says that male partners were selected from those who attend MCH clinics. This may be a limitation for the study as the recruitment is from a population of potentially more involved partners but this is not discussed in the limitations. We also may need more discussion on the translation from Swahili to English in the coding process as this is a potential issue in how meaning was potentially changed. 

Finally, there should also be potential discussion of a strengths for the study. As previously stated, this is a novel paper and such findings be stated.

Author Response

1.Overall this article offers a novel look into potential barriers to cervical cancer screenings in Central Kenya. The focus on the role of the male partner in attempting to increase cervical cancer screenings and other prevention methods is a fascinating project and, for this study, we see novel findings. While the results are interesting, I believe the total manuscript would benefit from additional revisions. The introduction, methods, and discussion could all be improved in various capacities. Introduction: Overall, it is simply too long of an introduction section. The total length is around 3 pages and must be cut. For example, lines 79-85 feel out of place in the introduction and would be better suited for the discussion. 

We have reviewed the introduction and excluded redundant information the introduction is now focused on 3 paragraphs.

2.On lines 98-113 the authors go into great detail for a single article for one justification for the study. This can be greatly reduced in size or moved to the discussion but as it stands, this is not suitable.

We have reviewed this and removed details in the introduction section.  

3.We see this in the following paragraph with lines 114-123. Great amounts of detail for one study. This should be adjusted. 

We note of this important consideration and excluded redundant information.

4.Authors should re-read the introduction to ensure the narrative is coherent and not using space to overexplain single articles.

We have reviewed this in the introduction.  

5.Finally, for the introduction, we do not receive information on why this should be a qualitative study outside of faint mention in lines 132-134. It would benefit the paper to expand on the rationale for the chosen methodology.

We have clarified this further in the introduction and now lines168-172 reads. “Despite the critical role male partners can play in alleviating the burden of cervical cancer, there is minimal information on their involvement in the screening and treatment process including challenges and opportunities. Understanding these perspectives through a qualitative study is crucial in providing contextual understanding of social dynamics, and culturally sensitive interventions that could promote male involvement in cervical cancer prevention and treatment”

6.Methods: On line 168 the researchers discussion that "couples" were selected but does not define how they determine who is classified as a couple. Please add.

We have clarified this in the method section lines 103-104 now reads “Couples defined by (men and women of reproductive age and in an intimate relationship)”

7.In the data analysis step, the way in which transcripts were coded and analyzed is discussed but we do not see how many reviewers were included. Please add. We see that in line 223-225 there were multiple reviewers involved but only at the end. Clarify.

We have clarified in the method section that 2 analysts (line 150) were involved in the coding and analysis

8.Same thing for section 2.9, please specify the number of reviewers and coders assisting in this project.

We have clarified in the method section that 2 analysts were involved in the coding and analysis.

9.Findings Section: Table 1 and 2 are nice to have but I believe they are unnecessary. The same information could be conveyed in an additional paragraph in section 3.1.

We thank the reviewer, we have deleted table 1 and 2 and summarized all the demographic information in a single table and provide the summary as advised in table 1.

10.Discussion: Currently, due to how long the introduction is, the discussion feels quite truncated. I do not believe much needs done as this can be adjusted with reducing the volume of the intro.

We have reviewed the introduction and discussion sections and included relevant details missing.

11.Limitations: In general, the limitations section is rather bland and could use refinement. The methods section is quite strong in some regards but could use a bit of expansion in a couple areas. For inclusion criteria, lines 175-176, the manuscript says that male partners were selected from those who attend MCH clinics. This may be a limitation for the study as the recruitment is from a population of potentially more involved partners, but this is not discussed in the limitations. We also may need more discussion on the translation from Swahili to English in the coding process as this is a potential issue in how meaning was potentially changed. Finally, there should also be potential discussion of a strengths for the study. As previously stated, this is a novel paper and such findings be stated.

We acknowledge this important note, we have clarified the study strength and limitation in the main document lines 454-459 now reads. “An important strength of our study is the use of qualitative approaches, which enabled a deeper and more nuanced understanding of male involvement in cervical cancer treatment and prevention. A key limitation, however, is the potential for social desirability bias due to the dynamics of focus group discussions. Additionally, our recruitment targeted couples accessing health services, and their perspectives may differ from those of other populations. Despite these limitations, the study provides an in-depth understanding of the barriers to, and opportunities for, male partner involvement in cervical cancer treatment and prevention”.

For the translation from Swahili to English no meaning was lost during transcription and coding process as the researcher and the assistants were both proficient with both languages and were able to interpret all the words verbalized.

Round 2

Reviewer 1 Report (Previous Reviewer 1)

Comments and Suggestions for Authors

The submitted version is shorter, more concise, and indeed more suitable for publication. However, as I mentioned earlier, descriptive studies of this type do not convince me, as they are inherently non-reproducible and subject to selection bias. The manuscript also lacks reference to similar studies conducted in other regions and countries. The reduction of material from version 1 to version 2 further raises the risk of fragmentation. Unfortunately, this article does not convince me.

Author Response

  1. The submitted version is shorter, more concise, and indeed more suitable for publication. However, as I mentioned earlier, descriptive studies of this type do not convince me, as they are inherently non-reproducible and subject to selection bias. The manuscript also lacks reference to similar studies conducted in other regions and countries. The reduction of material from version 1 to version 2 further raises the risk of fragmentation. Unfortunately, this article does not convince me.

Thank you for your insightful feedback and for highlighting key areas of concern regarding the methodology and scope of the manuscript. We respectfully acknowledge your reservations about the reproducibility and generalizability of qualitative descriptive studies. However, we would like to emphasize that qualitative research, particularly when grounded in interpretivist and constructivist paradigms as in this study serves a critical and complementary role in public health and health systems research. While not intended to produce statistically generalizable findings, qualitative studies offer rich, in-depth insights into human behavior, lived experiences, and social contexts that are often inaccessible through quantitative methods.

Regarding concerns about selection bias, we note that participant recruitment was purposive to ensure a diverse range of perspectives (including community members, health workers, and local leaders). This methodological rigor enhances credibility and transferability of findings across similar settings.

We agree with your point on the need to contextualize our findings within the broader literature. In response, we have now included in the discussion references 23 and 34 conducted in other regions. Line 375 now reads Notably, a study in Nigeria shown high willingness of men in supporting CC screening and prevention among their female partners which underscores the role of male involvement [23]. And line 402 reads Similarly, a qualitative study conducted in Uganda on involving men in CC prevention also found limited knowledge among men, with existing misinformation about risk factors and causes of cancer [35].

 Concern about fragmentation due to the reduction of material, the revised version was carefully condensed to meet journal and other reviewers recommendations. We addressed this while retaining the core themes and participant voices.

Reviewer 3 Report (Previous Reviewer 3)

Comments and Suggestions for Authors

Thank you for addressing my comments.

Author Response

Thank you for addressing my comments.

We are grateful for your time in reviewing our manuscript.

Reviewer 4 Report (New Reviewer)

Comments and Suggestions for Authors

No further comments. The revised manuscript addresses all prior comments. 

Author Response

No further comments. The revised manuscript addresses all prior comments.

We appreciate and we are very grateful for your time in reviewing our manuscript.

This manuscript is a resubmission of an earlier submission. The following is a list of the peer review reports and author responses from that submission.

Round 1

Reviewer 1 Report

Comments and Suggestions for Authors

I was presented with a paper for review titled:

"Challenges and Opportunities of Male Partner Involvement in Cervical Cancer Prevention and Control in Kenya: A Qualitative Analysis"

Below, I provide my comments on individual sections of the paper.

Abstract:

In my opinion, the abstract requires revision, as it is difficult to discern the article’s main focus based on its content.

The authors should specify the type of preventive engagement expected from men.

The abstract lacks punctuation in several places.

The abstract does not include numerical results, which should be a key feature of a scientific abstract.

The conclusion is overly general and would hold true even without conducting the study, making it devoid of meaningful insight.

Introduction:

The authors should clarify what characterizes male involvement in cervical cancer prevention. The entire introduction consists of general statements without addressing the core issue. These generalizations, incidentally, are very typical of large language models (LLMs).

Materials and Methods:

For me, a hallmark of scientific research is its reproducibility. Based on this section, I would not know how to conduct a similar study in my own country. In my view, this study is not reproducible for the reader.

Results:

Similar to the above, I have concerns about the study’s structure. The results are presented as individual fragments of conversations with participants. While this may be interesting, it lacks the characteristics of a scientific study.

I believe this format is unacceptable and unlikely to capture the interest of readers.

Author Response

Please find the authors' responses in the attachment.

Reviewer 2 Report

Comments and Suggestions for Authors

Dear Authors

Thanks for your practical and critical study. Cervical cancer is becoming one of the most important women's health problems, especially in low- and middle-income countries.

1-  In the introduction section, please mention the exact role of the male partner in MCH and its relationship with cervical cancer prevention.

2-How did you select the sample size?

3-How do the researchers address bias during data collection and interpretation?

4-The conclusion section appears long, it is better to summarize the conclusion.

5-What are the limitations and strengths of your study?

Author Response

(The authors gave the same response as above.)

Reviewer 3 Report

Comments and Suggestions for Authors
  • Findings repeat what has been widely studied in similar LMIC contexts
  • No new model, framework, or intervention is proposed
  • Eight challenges and six opportunities are discussed in excessive detail. Several themes are overlapping and repetitive (e.g., communication, support, time)
  • No reflexivity or clear explanation of researcher bias.
  • Unclear how data saturation was determined across diverse participant types.
  • Triangulation and trustworthiness are mentioned but not rigorously demonstrated.
  • Frequent grammar issues and inconsistent phrasing, eg “males spouses”

Author Response

(The authors gave the same response as above.)
